# GeneGrad: Gene-Specific Geometric Gradients for Cell Fate Prediction in Single-Cell Transcriptomics

**Shupeng Luxu**[1,2] **& Rong Ma**[1,2]
[1]Department of Biostatistics, Harvard T.H. Chan School of Public Health
[2]Broad Institute of MIT and Harvard
02115, Boston, MA, USA
{shupeng_luxu, rongma}@hsph.harvard.edu

## Abstract

Single-cell profiling technologies have enabled us to study the complex biological systems by resolving the gene expression variation on the single-cell level. Modeling cellular dynamics and elucidating their underlying regulatory mechanisms is essential to understand developmental processes, disease progression, and aging, and is beneficial to therapeutic strategies grounded in regenerative biology. While existing computational approaches infer cellular dynamics by reconstructing vector fields, they typically rely on gene-aggregated representations and primarily recover trajectory- or tree-like structures, limiting their ability to capture fine-grained, gene-specific, and geometrically diverse patterns.

Here, we introduce GeneGrad, a multi-scale geometric representation learning framework that models cellular dynamics through gene-specific gene expression gradients defined on the intrinsic cell-state manifold. By estimating local gradients for individual genes, GeneGrad captures gene-specific geometric patterns that are ignored by capturing the major variation. We validate GeneGrad on synthetic datasets with known geometric structure, demonstrating accurate gradient recovery and pattern identification. Applying GeneGrad to human induced pluripotent stem cell (iPSC) lung differentiation and hematopoietic progenitor datasets, we show that gene gradient geometry reveals early fate bias. Together, GeneGrad provides an interpretable and general framework for learning meaningful geometric representations of cellular dynamics from static single-cell data. We further extend the framework beyond single-cell transcriptomics to additional modalities, including ATAC-seq, highlighting its utility for uncovering biologically meaningful signals in complex cellular processes.

## 1 Introduction

Single-cell profiling technologies have fundamentally reshaped our understanding of cellular heterogeneity and dynamics by enabling the systematic measurement of gene expression at cellular resolution (Angerer et al.). Capturing cellular dynamics and elucidating the underlying regulatory mechanisms are essential for understanding developmental processes, disease progression, and aging (Saelens et al.). Identifying the key drivers of cell-state transitions and early fate bias can further enable cellular reprogramming and advance regenerative medicine. However, most of the single-cell measurements are only static snapshots, making the inference of cellular dynamics from high-dimensional, noisy observations a computational challenge (Weinreb et al. (b)).

A broad class of computational methods has been developed to infer cellular dynamics from scRNA-seq data, including pseudotime inference, RNA velocity (Bergen et al.), and vector-field reconstruction approaches. While these methods have been successful in recovering dominant variation trends, they rely primarily on global or gene-aggregated representations of change. RNA velocity methods, for example, infer a single velocity vector per cell by pooling information across genes under kinetic assumptions that may not hold in complex, non-equilibrium systems. More recent manifold-aware approaches reconstruct continuous vector fields and analyze their divergence or stability properties,

yet still prioritize global fields over gene-specific local structure (Qiu et al.). Consequently, these representations tend to emphasize trajectory- or tree-like dynamics and often ignore fine-grained geometric patterns that are critical for resolving early fate bifurcations, localized heterogeneity, and unstable progenitor states.

We posit that cell-state changes are governed not only by global trend, but also by local, gene-specific geometric signals defined on the intrinsic cell-state manifold. In particular, the direction of maximal expression increase for an individual gene within a cell's local neighborhood encodes biologically meaningful information about regulatory forces and fate bias that is lost under gene aggregation. Capturing such signals requires a local, per-gene, and geometry-aware representation.

To this end, we introduce **GeneGrad**, a multi-scale geometric representation learning framework for single-cell data that models cellular dynamics through gene-specific expression gradients. Gene-Grad:

- defines local gene expression gradients directly on the high-dimensional cell-state manifold, yielding a gene-specific geometric representation at single-cell resolution;

- projects these gradients into low-dimensional embeddings using a neighborhood-aware, geometry-consistent procedure that preserves directional structure; and

- analyzes the resulting gene-gradient vector fields to extract fate-relevant geometric patterns, including divergence, local heterogeneity, and directional alignment.

We validate GeneGrad on synthetic datasets with known geometric structure, demonstrating accurate recovery of diverse gradient patterns beyond simple trajectories. We further apply GeneGrad to the human induced pluripotent stem cell (iPSC) lung differentiation (Hurley et al.) and hematopoiesis dataset (Weinreb et al. (a)), where gene-gradient geometry reveals early fate bias. Together, these results position GeneGrad as a general and interpretable framework for learning meaningful geometric representations of cellular dynamics from static single-cell data.

## 2 BACKGROUND AND RELATED WORK

### 2.1 TRAJECTORY AND VELOCITY-BASED MODELS

Single-cell RNA sequencing (scRNA-seq) captures high-dimensional snapshots of cellular states but does not directly measure temporal evolution. To infer dynamics from static data, pseudotime methods order cells along a latent progression and recover coarse differentiation trajectories, but they remain descriptive and do not model directional change (Qu et al.; Li et al. (a)).

RNA velocity models estimate instantaneous change using unspliced/spliced RNA counts. Classical methods (Bergen et al.; Li et al. (b)) infer one velocity vector per cell by pooling signals across genes under global kinetic assumptions that can break in non-equilibrium settings. More recent variants relax parts of these assumptions, yet still rely on gene-aggregated signals to recover dominant trends. As a result, they emphasize large-scale trajectory-like structure and can obscure early, gene-specific cues of fate bias and lineage commitment.

### 2.2 VECTOR FIELD AND MANIFOLD-BASED DYNAMICS

Recent work reframes cellular dynamics as continuous vector fields on the cell-state manifold, enabling geometric analyses such as Jacobians (Qiu et al.), divergence, potential landscapes, or Hodge-theoretic decompositions (Maehara & Ohkawa). These tools provide a richer view than trajectories alone, but typically focus on a single global field that summarizes collective behavior across genes. Gene-level geometric signals, e.g., localized divergence near bifurcation points or heterogeneous regulatory forces within progenitors, may therefore be averaged out.

### 2.3 REPRESENTATION GAP

Overall, existing approaches predominantly produce global trajectories or gene-aggregated vector fields. What is missing is a ***per-gene, geometry-aware*** representation that quantifies, at each cell,

the direction and strength of maximal local expression increase for individual genes, while remaining compatible with downstream vector-field analysis and fate prediction. GeneGrad fills this gap by estimating gene-specific expression gradients on the intrinsic manifold and leveraging their geometry to predict cell fate.

## 3 PROBLEM FORMULATION AND OVERVIEW

**Inputs.** We consider a single-cell dataset consisting of $n$ cells and $G$ genes. Each cell $i$ is represented by a $d$-dimensional latent embedding $\mathbf{x}_i \in \mathbb{R}^d$, obtained from the original gene expression matrix using a dimensionality reduction method such as PCA. For each gene $g$, we denote the expression level at cell $i$ by $y_{ig} \in \mathbb{R}$. We assume that the latent embeddings capture the intrinsic geometry of the cell-state manifold, and that local neighborhoods in this space reflect meaningful biological similarity.

**Outputs.** Our goal is to learn a gene-specific geometric representation of cellular dynamics. Specifically, for each gene $g$ and cell $i$, we seek to estimate a local gradient vector

$$\nabla y_g(\mathbf{x}_i) \in \mathbb{R}^d,$$

which encodes the direction and magnitude of the maximal local increase in gene expression with respect to the intrinsic coordinates of the cell-state manifold. Collectively, these gradients define a gene-specific vector field over cells, providing a local and interpretable representation of gene-level dynamics.

**Problem Statement.** Given high-dimensional cell embeddings $\{\mathbf{x}_i\}_{i=1}^n$ and gene expression measurements $\{y_{ig}\}$, we aim to infer gene-specific local gradient fields that (i) respect the local geometry of the cell-state manifold, (ii) preserve directional consistency when mapped to low-dimensional embeddings, and (iii) can be analyzed to reveal fate-relevant geometric patterns.

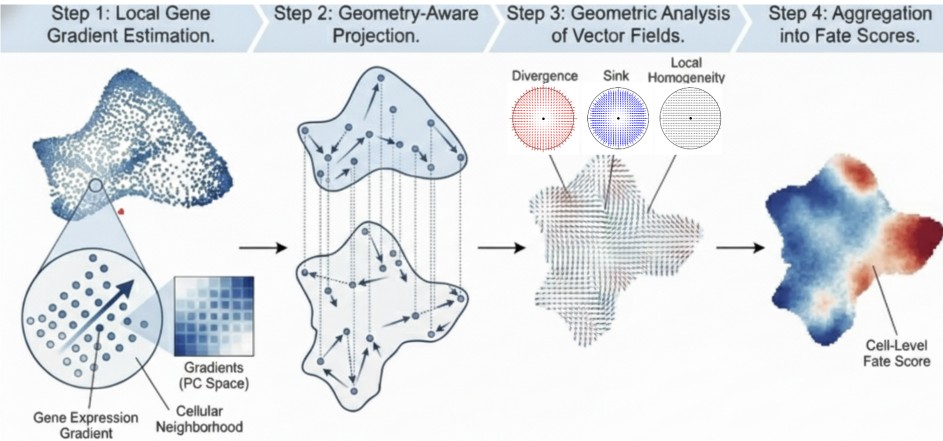

Figure 1: **GeneGrad Flowchart.** GeneGrad consists of four stages: (1) local gene gradient estimation in high-dimensional PC space, (2) geometry-aware projection to low-dimensional embeddings, (3) geometric analysis of gene-gradient vector fields, and (4) aggregation into cell-level fate scores.

**Overview of GeneGrad.** GeneGrad addresses this problem through four stages, illustrated in Fig-1. First, we estimate local gene expression gradients directly in the high-dimensional latent space using neighborhood-based models, yielding gene-specific representations at single-cell resolution. Second, we transfer these gradients to low-dimensional embedding spaces (e.g., UMAP or PHATE) using a geometry-aware projection that preserves local directional alignment. Third, we analyze the resulting gene-gradient vector fields using a set of geometric pattern, including sink–source behavior, directional homogeneity, and spatial autocorrelation, to characterize local and global structure. Finally, we aggregate gene-level gradient information to compute cell-level scores that summarize fate-relevant geometric signals and enable downstream cell fate analysis.

## 4 METHODOLOGY

GeneGrad is a geometric representation learning framework that models cellular dynamics through gene-specific expression gradients defined on the intrinsic cell-state manifold. An overview of the framework is shown in Fig-1.

### 4.1 LOCAL GENE GRADIENT ESTIMATION IN HIGH DIMENSIONS

Each cell $i$ is represented by a latent embedding $\mathbf{x}_i \in \mathbb{R}^d$ (e.g., PCA) and a gene expression value $y_i \in \mathbb{R}$ for a given gene. Local neighborhoods $\mathcal{N}(i)$ are defined in the latent space.

For each neighboring cell $j \in \mathcal{N}(i)$, we define relative displacements $\mathbf{x}'_j = \mathbf{x}_j - \mathbf{x}_i$ and $y'_j = y_j - y_i$. We estimate a local gene expression gradient $\mathbf{w}_i \in \mathbb{R}^d$ by fitting a regularized linear model:

$$\min_{\mathbf{w}} \sum_{j \in \mathcal{N}(i)} \left( y'_j - \mathbf{x}_j'^\top \mathbf{w} \right)^2 + \alpha \|\mathbf{w}\|_2^2.$$

This locally linear approximation captures the direction and magnitude of maximal local expression increase for each gene at the single-cell level resolution. Unless otherwise stated, we use ridge regression for its stability and computational efficiency.

### 4.2 GEOMETRY-AWARE PROJECTION TO LOW-DIMENSIONAL EMBEDDINGS

Gene gradients are learned in a high-dimensional latent space, while interpretation and visualization are typically performed in two-dimensional embeddings (e.g., UMAP). Rather than directly projecting gradient vectors, we reconstruct directionally consistent representations that respect the local manifold geometry.

Let $E(\mathbf{x}_i) \in \mathbb{R}^2$ denote the embedding of cell $i$. For each neighbor $q \in \mathcal{N}(i)$, we compute the alignment between the high-dimensional gradient $\nabla f_i$ and the local displacement $d_{iq} = \mathbf{x}_q - \mathbf{x}_i$:

$$r_{iq} = \frac{\langle d_{iq}, \nabla f_i \rangle}{\|d_{iq}\| \, \|\nabla f_i\|}.$$

Alignment scores are converted into normalized weights:

$$\alpha_{iq} = \frac{\exp(r_{iq}/\sigma)}{\sum_{q' \in \mathcal{N}(i)} \exp(r_{iq'}/\sigma)}.$$

The projected gradient in embedding space is reconstructed as

$$\mathbf{v}_i = \sum_{q \in \mathcal{N}(i)} \alpha_{iq} \big( E(\mathbf{x}_q) - E(\mathbf{x}_i) \big).$$

This neighborhood-aligned reconstruction preserves relative directional structure between gene gradients and the local manifold.

### 4.3 GENE-GRADIENT VECTOR FIELD GEOMETRY ANALYSIS

Projected gene gradients form vector fields over cells. We characterize their geometric structure using complementary metrics:

- Sink–source behavior, measuring whether gradients act as local sources or sinks relative to nearby cells;
- Local directional homogeneity, quantifying alignment among neighboring gradient vectors;
- Spatial autocorrelation, assessing whether gradient-derived quantities are spatially organized in the embedding space;
- Global directional alignment, summarized by the dominance of the leading singular vector of the gradient field.

Together, these measures capture both local heterogeneity and global organization of gene-specific geometric signals. The detailed definition of geometric patterns is in Appendix D.

## 4.4 Cell-level Fate Scoring

We interpret cell fate as an emergent property of aggregated gene-level geometric signals. For each gene $g$ and cell $i$, we compute the magnitude of the projected gradient $\mathbf{v}_{ig}$ as

$$m_{ig} = \|\mathbf{v}_{ig}\|_2.$$

Cell-level scores are obtained by aggregating across genes:

$$\bar{m}_i = \frac{1}{G} \sum_{g=1}^{G} m_{ig},$$

where $G$ is the number of genes. These scores summarize the strength of fate-relevant geometric signals and are used for downstream cell fate prediction and analysis.

## 5 Experiments

### 5.1 Validation and Benchmark on Synthetic Dataset

#### 5.1.1 Benchmark Design

To evaluate whether GeneGrad can recover gene-specific geometric structures under controlled conditions, we constructed a synthetic benchmark with a known ground-truth manifold. We sampled cells from a linear band manifold embedded in high-dimensional space ($d = 40$), characterized by a dominant axial direction, limited transverse variation, and isotropic Gaussian noise. Full details of data generation and ground-truth geometry are provided in Appendix A.

On this manifold, we designed two gene expression patterns, an axial pattern and a radial pattern (Fig. 2), serving as proxies for linear developmental progression and fate bifurcation, respectively. In the axial pattern, gene expression varies primarily along the dominant manifold direction, representing a globally aligned gradient that is coherent across the cell population. In contrast, the radial pattern exhibits symmetric expression increases away from a central axis, mimicking transcriptomic divergence from a progenitor state and capturing the local geometric structure associated with developmental branch points.

To our knowledge, GeneGrad is the first method to estimate gene-specific expression gradients directly in high-dimensional latent space. We therefore benchmarked it against a $k$-nearest neighbor (KNN) baseline using analytical ground truth. We quantitatively assessed performance at two complementary levels:

1. **Local Gradient Accuracy**: The precision of individual cell-wise vector directions.

2. **Global Geometric Recovery**: The coherence of the vector field, characterized by the flux (divergence) of the gradient field.

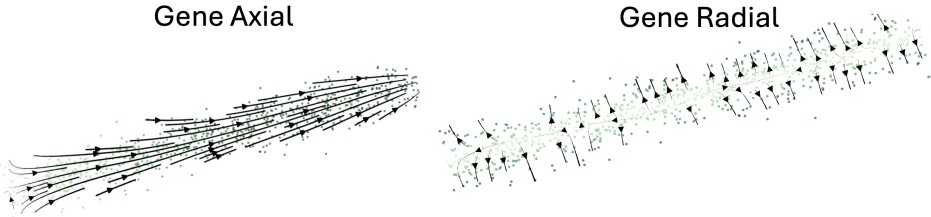

Gene Axial            Gene Radial

Figure 2: **Synthetic data with structured geometric patterns.** The streamlines indicate gene expression gradients estimated by GeneGrad. Dots represent individual cells and are colored by gene expression level, with darker colors indicating gene expression.

### 5.1.2 BENCHMARK 1: GRADIENT ESTIMATION BENCHMARK

We evaluated gradient estimation accuracy by comparing GeneGrad with a KNN-based baseline across different neighborhood sizes $k$ on synthetic data with known ground-truth geometry. We focused on two representative gene expression patterns defined on the same underlying manifold: an axial pattern and a radial pattern, which capture globally aligned and locally divergent geometric structures, respectively.

Gradient estimation accuracy was quantified using cosine similarity between the estimated gene expression gradient and the ground-truth gradient at each cell. This metric measures directional agreement independent of magnitude and is therefore well suited for assessing whether the orientation of local gene expression change is correctly recovered.

For the axial pattern, both GeneGrad and the KNN baseline achieve consistently high cosine similarity across all tested neighborhood sizes (Table 1). In this regime, the KNN baseline slightly outperforms GeneGrad, particularly at smaller $k$. The performance of GeneGrad improves significantly as the $k$ increases.

In contrast, the radial pattern reveals a different behavior. While the KNN baseline exhibits better performance at smaller neighborhood sizes ($k \leq 50$), it shows a rapid decline in cosine similarity as $k$ increases, indicating substantial oversmoothing of locally varying gradient directions. GeneGrad, by explicitly modeling gene-specific local geometry, remains significantly more robust to increasing neighborhood size and consistently achieves higher cosine similarity for larger values of $k$ ($k \geq 50$).

Importantly, radially structured gene expression patterns are both biologically more relevant and more prevalent in real single-cell systems, as they reflect fate commitment and transcriptomic divergence from progenitor states during developmental branching. Moreover, in the GeneGrad setting, neighborhood sizes are typically chosen to be moderately large (often $k \geq 50$) to ensure numerical stability and robustness to noise. In this biologically and practically relevant regime, GeneGrad outperforms the KNN baseline, highlighting its advantage in accurately recovering gene-specific gradients under realistic analysis conditions.

| Gene Pattern | $k$ | GeneGrad | KNN |
|---|---|---|---|
| Gene Axial | 30 | 0.891 | 0.978 |
| | 50 | 0.917 | 0.995 |
| | 70 | 0.927 | 0.997 |
| | 100 | 0.986 | 0.998 |
| | 120 | 0.986 | 0.998 |
| | 150 | 0.986 | 0.999 |
| Gene Radial | 30 | 0.796 | 0.903 |
| | 50 | 0.775 | 0.807 |
| | 70 | 0.763 | 0.670 |
| | 100 | 0.653 | 0.513 |
| | 120 | 0.603 | 0.403 |
| | 150 | 0.417 | 0.275 |

Table 1: **Gradient-level benchmarking results (cosine similarity) on selected neighborhood sizes** $k$**.** We report cosine similarity between the estimated gradients and the ground-truth gradients for GeneGrad and the KNN baseline across different synthetic gene patterns.

### 5.1.3 BENCHMARK 2: GEOMETRIC PATTERN RECOVERY VIA FLUX ESTIMATION

Beyond local gradient estimation, we evaluate the recovery of higher-order geometric patterns by comparing flux (divergence/sink) induced by the estimated vector fields with analytical ground truth. Flux correlation measures the agreement between the flux structure of an estimated field and the ground-truth vector field, and thus serves as a field-level metric for geometric pattern recovery.

For axially structured patterns, flux comparison with the ground truth is not informative, becasue the axial pattern corresponds to a globally coherent and approximately zero-flux continuous flow field.

In contrast, radial patterns induce spatially varying flux that is highly sensitive to local geometric structure, making flux estimation a discriminative benchmark. As shown in Table 2, while the KNN baseline achieves higher flux correlation at small neighborhood sizes, its performance rapidly deteriorates as $k$ increases, eventually yielding near-zero or negative correlation. This degradation arises from excessive smoothing across heterogeneous local directions, which collapses the underlying divergence structure. GeneGrad, by explicitly modeling gene-specific local geometry, preserves directional heterogeneity and maintains consistently high flux correlation, substantially outperforming the KNN baseline at larger neighborhood sizes.

Importantly, in practical single-cell analyses, neighborhood sizes are typically chosen to be moderately large (often $k \geq 50$) to reduce noise and improve numerical stability. In this biologically and practically relevant regime, GeneGrad demonstrates clear advantages over the KNN baseline, highlighting its utility for recovering meaningful geometric patterns from noisy, high-dimensional single-cell data.

| $k$ | GeneGrad | KNN Baseline |
|-----|----------|--------------|
| 30 | 0.781 | 0.923 |
| 50 | 0.775 | 0.853 |
| 70 | 0.772 | 0.685 |
| 100 | 0.757 | 0.332 |
| 120 | 0.767 | 0.117 |
| 150 | 0.754 | -0.087 |

Table 2: **Flux-level benchmarking (flux correlation) results on selected neighborhood sizes $k$.** Comparison of Flux Correlation with Ground Truth between GeneGrad and the KNN Baseline across Different Neighborhood Sizes $k$.

## 5.2 Validation and Benchmark on Real Biological Dataset

After validating GeneGrad on synthetic datasets with known ground-truth gradients, we applied it to developmental single-cell RNA-seq systems, including: (1) human iPSC lung differentiation (Hurley et al., and (2) hematopoiesis (Weinreb et al. (a)). In these systems, initially undifferentiated progenitor cells commit to one of two distinct terminal fates. For each cell, we computed a cell-wise gradient score and used it to estimate the fate bias of the progenitor populations.

### 5.2.1 Benchmark: Cell Fate Prediction

Because ground-truth fate outcomes in these systems are available only at clonal resolution, we compared the fate bias inferred by GeneGrad with that obtained using CoSpar (Wang et al.), a well-established cell fate prediction method using single-cell RNA-seq and lineage-tracing information. We observed that the progenitor fate bias estimates produced by GeneGrad are highly correlated with those from CoSpar (iPSC, $r = \mathbf{0.889}$; hematopoiesis, $r = \mathbf{0.857}$). The cell fate bias estimated by GeneGrad and CoSpar are shown in Figure 3, indicating that GeneGrad effectively captures fate-relevant directional signals within gene expression principal component space.

Notably, GeneGrad outperforms CoSpar in terms of general applicability, as it does not require lineage-tracing information as input. While lineage-based approaches rely on specialized experimental protocols that are often inaccessible or less common than standard single-cell technologies, GeneGrad operates solely on scRNA-seq and fully utilizes the geometric information of the cellular manifold to provide reliable cell fate prediction.

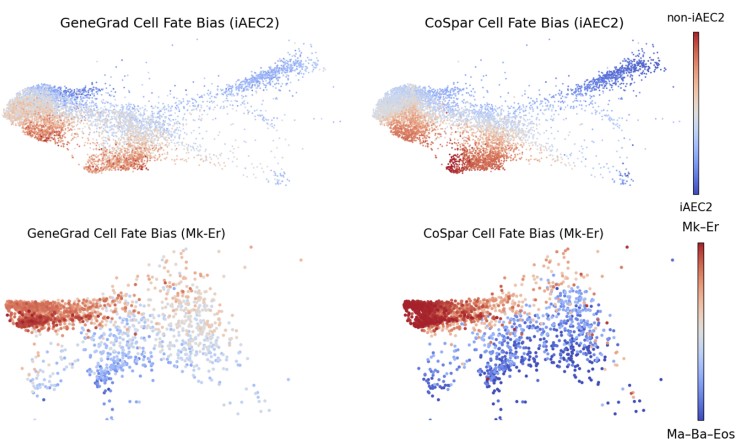

Figure 3: **Cell fate prediction result by GeneGrad and CoSpar.** GeneGrad-derived cell fate bias is compared with lineage-based CoSpar results for human iPSC lung differentiation (top) and hematopoiesis (bottom). Despite relying solely on scRNA-seq data, GeneGrad shows strong agreement with CoSpar across progenitor populations (iPSC: Spearman Correlation $r = $ **0.889**; hematopoiesis: $r = $ **0.857**), indicating that it captures fate-relevant directional signals without requiring lineage information.

## 6   CONCLUSION

In this work, we introduced **GeneGrad**, a geometric representation learning framework that models cellular dynamics through gene-specific expression gradients defined on the intrinsic cell-state manifold in gene expression space. By estimating local gene expression gradients within each cell's neighborhood and analyzing their geometric organization, GeneGrad provides a multi-scale representation of transcriptional change and enables the characterization of per-gene dynamics on the cellular manifold.

Through benchmarking on synthetic datasets, we demonstrated that GeneGrad accurately recovers both local gene expression gradients and higher-order geometric patterns associated with fate bifurcation. As GeneGrad is, to our knowledge, the first method for per-gene expression gradient estimation on the cell-state manifold, we compared it against a $k$-NN-based baseline. GeneGrad consistently outperformed the baseline in recovering both ground-truth gradient directions and global geometric structure.

We further evaluated GeneGrad on real biological systems, including human iPSC lung differentiation and hematopoiesis. In these datasets, GeneGrad-derived progenitor fate bias estimates show strong agreement with predictions from CoSpar, a well-established cell fate inference method that integrates scRNA-seq and lineage-tracing information. Notably, GeneGrad achieves comparable performance using only scRNA-seq data, without relying on additional experimental modalities, thereby substantially improving the general applicability of the approach.

Overall, GeneGrad provides a biologically meaningful and generalizable representation of single-cell dynamics that enables the estimation of progenitor cell fate from standard single-cell data. As such, GeneGrad offers a powerful tool for studying cell fate commitment and developmental dynamics, and more broadly, processes such as disease progression and aging. In future work, we plan to extend GeneGrad to additional biological settings, including tumor evolution and perturbation-induced cell-state changes, as well as to other modalities such as scATAC-seq and proteomics, leveraging multimodal data to learn more informative biological representations.

MEANINGFULNESS STATEMENT

A meaningful representation of life should capture not only cellular states, but also the latent forces that govern how cells change and commit to fates. In this work, we view cellular dynamics as a geometric phenomenon on the intrinsic cell-state manifold and propose gene-specific expression gradients as a meaningful representation of these forces. By resolving local, gene-level geometric signals rather than global trajectories, GeneGrad reveals fate bias, instability, and heterogeneity that are otherwise obscured. This geometric perspective enables interpretable reasoning about cell fate from static data and provides a general framework for linking molecular regulation to emergent developmental dynamics.

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

# A SYNTHETIC DATA GENERATION

## A.1 LINEAR BAND MANIFOLD CONSTRUCTION

We construct a synthetic dataset in which cells lie on a linear band manifold with known intrinsic geometry. For each cell $i$, we sample an axial coordinate

$$u_i \sim \text{Uniform}\left(-\frac{L}{2}, \frac{L}{2}\right)$$

and a transverse coordinate

$$v_i \sim \mathcal{N}(0, \sigma^2),$$

where $L$ controls band length and $\sigma$ controls band width. The two-dimensional coordinates are optionally rotated by an angle $\theta$:

$$\mathbf{x}_i = (u_i, v_i)\mathbf{R}^\top(\theta).$$

## A.2 HIGH-DIMENSIONAL EMBEDDING

To embed the band into a higher-dimensional space, we construct $\mathbf{z}_i \in \mathbb{R}^D$ by setting

$$\mathbf{z}_i^{(1:2)} = \mathbf{x}_i, \quad \mathbf{z}_i^{(3:D)} \sim \mathcal{N}(0, \epsilon^2),$$

where $\epsilon$ controls isotropic noise in orthogonal dimensions. This yields a linear manifold embedded in high-dimensional space with controlled noise.

## A.3 GENE EXPRESSION SIMULATION

We simulate three genes with distinct geometric dependencies:

**Axial gene:**

$$g_i^{\text{axial}} = \text{Scale}(u_i),$$

monotonically varying along the principal axis.

**Radial gene:**

$$g_i^{\text{radial}} = \text{Scale}(|v_i|),$$

symmetric about the band center.

**Transverse gene:** The axial coordinate is partitioned into $K$ bins, and within each bin, expression increases monotonically with $v$:

$$g_i^{\text{trans}} = \text{Scale}\big(\text{Rank}_{\text{bin}(u_i)}(v_i)\big).$$

Expression values are discretized to integer counts in $[0, 20]$ to simulate sequencing-like measurements.

# B ALTERNATIVE LOCAL GRADIENT ESTIMATORS

We evaluated several alternative estimators for local gene gradients, including Poisson and negative binomial generalized linear models, as well as rank-based neural networks. While these models can better accommodate count-based noise, they incur higher computational cost and did not consistently improve alignment with ground-truth gradients on synthetic benchmarks.

Across all datasets, ridge regression achieved the highest average correlation between projected gradient directions and expression variation, while remaining numerically stable and scalable. Therefore, ridge regression is used throughout the main experiments.

## C   ROBUSTNESS HEURISTICS

To improve the stability of local gradient estimation in sparse or noisy settings, we apply the following heuristics:

- neighbors with zero gene expression are excluded from local regression;
- neighbors with identical expression values ($y_j = y_i$) are filtered to avoid degenerate gradients;
- if more than a fixed fraction of neighbors exhibit zero expression, the gradient is set to zero.

These steps reduce spurious gradient directions caused by dropouts and ensure robust estimation without introducing additional modeling assumptions.

## D   GENE-GRADIENT VECTOR FIELD PATTERN ANALYSIS

GeneGrad represents gene-specific dynamics as vector fields defined over cells in a low-dimensional embedding space. Here we provide formal definitions of the geometric metrics used to characterize local and global structure of these vector fields.

### D.1   SINK–SOURCE BEHAVIOR

Let $\mathbf{v}_i \in \mathbb{R}^2$ denote the projected gradient vector of a gene at cell $i$, and let $\mathcal{N}_k(i)$ denote the set of $k$ nearest neighbors of $i$ in embedding space. We define a sink–source score that measures whether gradients point toward or away from neighboring cells:

$$\text{SS}_i = \frac{1}{|\mathcal{N}_k(i)|} \sum_{j \in \mathcal{N}_k(i)} \frac{(\mathbf{y}_i - \mathbf{y}_j)^\top \mathbf{v}_i}{\|\mathbf{y}_i - \mathbf{y}_j\| \, \|\mathbf{v}_i\|}.$$

Positive values indicate source-like behavior (outward flow), while negative values indicate sink-like behavior. Cells with near-zero gradient magnitude are excluded to avoid numerical instability.

### D.2   LOCAL DIRECTIONAL HOMOGENEITY

To quantify local coherence of gradient directions, we normalize each projected gradient to unit length:

$$\hat{\mathbf{v}}_i = \frac{\mathbf{v}_i}{\|\mathbf{v}_i\|}.$$

Local directional homogeneity is defined as the average cosine similarity between a cell's gradient and those of its neighbors:

$$H_i = \frac{1}{|\mathcal{N}_k(i)|} \sum_{j \in \mathcal{N}_k(i)} \hat{\mathbf{v}}_i^\top \hat{\mathbf{v}}_j.$$

Higher values indicate locally aligned vector fields, while low or negative values suggest heterogeneous or unstable local geometry.

### D.3   SPATIAL AUTOCORRELATION

To assess whether scalar quantities derived from gene-gradient fields (e.g., gradient magnitude or sink–source score) exhibit spatial structure, we compute Moran's I statistic. Let $z_i$ denote a standardized scalar value at cell $i$, and let $\mathbf{W}$ be a row-normalized $k$NN weight matrix. The global Moran's I is defined as:

$$I_{\text{global}} = \frac{n}{S_0} \frac{\sum_i z_i \sum_j W_{ij} z_j}{\sum_i z_i^2}, \quad S_0 = \sum_{i,j} W_{ij}.$$

Local Moran's I values are computed analogously for each cell. Statistical significance is assessed using permutation testing.

### D.4 GLOBAL DIRECTIONAL ALIGNMENT

To quantify the presence of a dominant global direction in a gene's gradient field, we perform singular value decomposition on the centered gradient matrix. Let $\mathbf{V} \in \mathbb{R}^{n \times 2}$ denote the matrix of projected gradients after mean-centering. We compute:

$$\mathbf{V} = \mathbf{U}\mathbf{\Sigma}\mathbf{R}^\top,$$

where $\mathbf{\Sigma} = \mathrm{diag}(\sigma_1, \sigma_2)$. We define the global alignment score as:

$$\rho = \frac{\sigma_1}{\sigma_1 + \sigma_2}.$$

Values close to 1 indicate a strongly aligned global flow, whereas values near 0.5 indicate isotropic or heterogeneous directional structure.

### D.5 RELATION TO CELL FATE ANALYSIS

The above metrics characterize complementary aspects of gene-gradient geometry. In practice, genes associated with fate commitment exhibit strong sink–source behavior, high spatial autocorrelation, and increasing directional homogeneity near bifurcation regions. These geometric signatures are subsequently aggregated at the cell level to derive fate-relevant scores, as described in the main text.

