# OpenReview forum: "GeneGrad: Gene-Specific Geometric Gradients for Cell Fate Prediction in Single-Cell Transcriptomics"
_ICLR.cc/2026/Workshop/LMRL — ICLR 2026 Workshop LMRL Poster_

### Official Review · Reviewer_jmeZ · 2026-02-10
**Gene-specific gradients for fate prediction: positioning and benchmarking concerns**

**Rating:** 4
**Confidence:** 4

**Review:**

This paper proposes GeneGrad, a method for estimating gene-specific local expression gradients on the cell-state manifold and using their geometric organization for fate prediction. The focus on per-gene directional structure rather than gene-aggregated dynamics is well motivated, and the synthetic experiments demonstrate internal consistency of the proposed estimator. However, I have several concerns regarding positioning and evaluation.

1. Lack of comparison to existing work. There are several existing methods that perform gene-level dynamical inference. For example, scEGOT and OTVelo use optimal transport to estimate gene-specific local directional changes in expression and explicitly discuss extensions to multimodal data. There are also splicing-free RNA velocity methods (e.g., TFVelo, noSpliceVelo). In this context, the claim that GeneGrad is the first method to estimate gene-specific expression changes in high-dimensional latent space appears overstated.

2. Limited benchmarking. The benchmarking for cell fate prediction is limited and compares primarily to CoSpar, which relies on lineage tracing. However, many velocity- and OT-based methods produce transition matrices or kernels that can be directly passed to CellRank for fate estimation. Such baselines should be included.

3. Reliance on low-dimensional embeddings. While gradients are estimated in PCA space, the geometric analysis and visualization rely on UMAP or PHATE, which can be sensitive to parameter choices. Clarifying the robustness of the results to the choice of embedding would strengthen the paper.

Overall, the proposed framework is promising, but the evaluation and positioning are incomplete. Addressing these points, or clearly articulating why such comparisons are not appropriate, would substantially strengthen the work.

---

### Official Review · Reviewer_1qNK · 2026-02-24
**The paper introduces a geometry-aware method that estimates per-cell, per-gene local gradient vector fields in latent space. The idea is novel and the framework is simple, but the evaluation is limited, and some components lack theoretical justification.**

**Rating:** 6
**Confidence:** 4

**Review:**

Pros:

1. The idea is novel in the sense that no prior method explicitly estimates per-cell, per-gene gradient vector fields. It uses the local gradient, which gives us local directional information which is a stronger than a smoothness score.

2. The paper includes both synthetic validation and real biological applications. The authors assess higher-order geometric structure via flux-like metrics, which provides quantitative grounding. And the method works well on real data, as it has a high correlation with CoSpar.

3. Showing that this model can scale to larger neighborhood sizes is valuable. GeneGrad maintains high accuracy and flux correlation at large k values, while maintaining per-gene representations which assuages one of my concerns about over smoothing.

4. The method is computationally transparent and reproducible, as all of the core components are well understood.

Cons:

1. Comparing the gradient estimation only to a standard kNN baseline on synthetic data is a weak empirical benchmark. The authors should at least compare against CellRank 2 [2], and CytoTRACE 2 [3] or CytoTRACE[1]. Additionally, one important feature that they note is that Genegrad can predict progenitor cell fate bias using only scRNA-seq data and compare it against CoSpar which acts as an oracle. They prove it correlates well with CoSpar but they fail to compare against any other baseline, which makes it hard to tell if this is a useful tool to researchers, as another method may work better.

2. UMAP is not isometric, and it does not preserve distances, angles, or local metric structures exactly. Cosine similarity between vectors depends on the inner product and if the embedding distorts angles, then cosine similarity will change under nonlinear transformation. UMAP is known to distort global and intermediate-scale geometry [4][5], which means that the directional homogeneity scores computed in 2D are not guaranteed to reflect true alignment structure in PCA space. It would be nice to test whether divergence measured in PCA space matches divergence in UMAP, and geometric patterns are stable across different embeddings.

3. Because GeneGrad estimates a local gradient at each cell by fitting a linear model in PCA space, it has to assume the manifold is locally linear, and neighbor displacements lie approximately in a flat tangent space. If the manifold is curved, then the resulting sink-source or flux patterns may incorporate bias and no longer model biological structure as well. It would be nice to understand how well curvature affects its gradient estimator. This does not appear to be an issue in the synthetic example the authors show (although they only provide a linear band which yields a flat manifold) and the real dataset, but it could appear in other datasets.

4. The authors note an application with ATAC-seq in the abstract but then never perform any analyses using it. This should be removed from the abstract.

5. It would be nice to see other results, given that this is a per-gene method. Perhaps specific genes with strong divergence patterns, or examples where GeneGrad reveals an interesting feature of specific genes?

6. How slow / fast is this method? If it is computing a local regression for every gene and every cell that could add up?

7. In D.4 I don't understand the global alignment score in the sense that it uses singular values instead of squared values (variance).

$$ \space $$
Overall for a workshop paper I think this is a weak accept. I really don't like that there is only 1 baseline tested, but overall the paper is well written and invokes an interesting discussion on a gene-level analyses.


$$ \space $$
Citations

[1] Gunsagar S. Gulati et al. ,Single-cell transcriptional diversity is a hallmark of developmental potential.Science367,405-411(2020).DOI:10.1126/science.aax0249

[2] Weiler P, Lange M, Klein M, Pe'er D, Theis F. CellRank 2: unified fate mapping in multiview single-cell data. Nat Methods. 2024 Jul;21(7):1196-1205. doi: 10.1038/s41592-024-02303-9. Epub 2024 Jun 13. PMID: 38871986; PMCID: PMC11239496.

[3] Kang, M., Gulati, G.S., Brown, E.L. et al. Improved reconstruction of single-cell developmental potential with CytoTRACE 2. Nat Methods 22, 2258–2263 (2025). https://doi.org/10.1038/s41592-025-02857-2

[4] Kobak, D., Berens, P. The art of using t-SNE for single-cell transcriptomics. Nat Commun 10, 5416 (2019). https://doi.org/10.1038/s41467-019-13056-x

[5] Arvanitidis, Georgios, Lars Kai Hansen, and Søren Hauberg. "Latent Space Oddity: on the Curvature of Deep Generative Models." International Conference on Learning Representations. 2018.

---

### Official Review · Reviewer_5BLh · 2026-02-24
**interesting idea, not sure of the practical application**

**Rating:** 6
**Confidence:** 2

**Review:**

This work introduce a method to compute gene-specific expression gradients. I thought the idea was interesting and I am hopeful that the authors can find more practical applications.

I still do not understand the point of computing gene-specific expression gradients. For the biological dataset, I don't understand how the validation was performed, did the authors had to select a particular gene related to the cell fate? Just naively, why can't we just compute some aggregate expression score across multiple marker genes, as a measure of cell fate bias?

For future work, I hope the authors can better explain and demonstrate what GeneGrad can do well compare to other methods.

---

### Meta-Review · Area_Chair_VF69 · 2026-02-25

**Recommendation:** Accept (Poster)
**Confidence:** 4

**Metareview:**

Accept.

---

### Decision · Program_Chairs · 2026-03-02

**Decision:**

Accept (Poster)

**Comment:**

Please see the meta-review.